# Validation of Multi-Frequency Inductive-Loop Measurement System for Parameters of Moving Vehicle Based on Laboratory Model

**DOI:** 10.3390/s24227244

**Published:** 2024-11-13

**Authors:** Zbigniew Marszalek, Krzysztof Duda

**Affiliations:** Department of Measurement and Electronics, AGH University of Krakow, 30 Mickiewicz Avenue, 30-059 Krakow, Poland; kduda@agh.edu.pl

**Keywords:** slim inductive-loop sensor, multi-frequency impedance measurement, vehicle magnetic profile (VMP), laboratory test bed, vehicle parameters, speed, wheelbase, vehicle length, vehicle axle, front overhang, rear overhang

## Abstract

The paper presents research on a system for measuring the parameters of a vehicle in motion and the process of validating it under laboratory conditions. The measurement system uses four inductive-loop (IL) sensors, two slim ILs and two wide ILs. The vehicle speed, wheelbase, length, and overhangs are all determined on the basis of a vehicle magnetic profile (VMP) waveform. VMPs are captured from the continuous IL-based impedance measurement. The impedance measurement for a single IL is performed simultaneously at three carrier frequencies. The uncontrolled measurement conditions in the field on a real road test bed (RTB), including the speed of passing vehicles, motivated the development of a laboratory test bed (LTB). This LTB serves as a model of an existing measurement setup installed on the road, i.e., the RTB. The LTB includes IL sensors and a movable model of the vehicle made in 1:50 scale. The LTB enables validation of the whole measurement system in the vehicle speed range from 10 km/h up to 150 km/h in 5 km/h increments in fully controlled conditions. The measurement results are presented in the distance domain, calculated from the VMPs and the measured speed. The largest errors in estimating vehicle-model body parameters, on a natural scale, do not exceed 4.3 cm.

## 1. Introduction

The measurement of road traffic parameters provides important information that can be utilized for planning infrastructure development, such as new roads or the reconstruction and reorganization of traffic and intersections in response to changing usage conditions. In order to carry out these measurements, automatic road traffic parameters measurement systems are increasingly used [1,2,3]. These systems pose a number of difficult requirements related to meteorological properties, immunity to interference and the technology of production and installation [4,5,6,7]. For continuous operation, the entire measurement process needs to be automated [8]. Uninterrupted measurements over a long period of time (months and years) require the measurement system to be maintenance-free, error-resilient and to have the longest possible operating time which is mainly limited by the lifetime of the applied sensors. Moreover, due to year-round use, the system must be immune to weather conditions. Systems for measuring the parameters of the vehicle in motion should be thoroughly tested. Field, i.e., road testing is time-consuming, and cannot be sufficiently controlled. For that reason, we propose to carry out all required tests in a laboratory in a fully controlled environment. This, in turn, requires the development of dedicated equipment, including scaled models of applied sensors and vehicles.

Road traffic parameters include, e.g., the measurements of vehicle speed, flow, and type for the purpose of classification. Continuous recording of these parameters results in statistical data, which are the most important traffic description volumes [9]. One of the most commonly used types of sensors for these systems are inductive-loop (IL) sensors because of their advantages, i.e., low price and high durability [10,11] as compared to other solutions. IL sensors are installed on the road surface which protects them against weather conditions and mechanical damage. This ensures trouble-free operation for several years. The installation process is not complicated and quick, which, combined with the low price of the sensor, is a key advantage that contributes to the popularity of IL sensors [12].

Initially, IL sensors were used only to detect the presence of a vehicle at a certain point on the road and to measure its speed [13]. However, a set of IL sensors is capable of providing much more data about a vehicle, such as the number of vehicle axles, along with their spacing [14], the total length of the vehicle [15,16], the length of overhangs or even the suspension height [17]. This is especially possible when a multi-frequency impedance measurement (MFIM) system for IL sensors is used [18,19,20,21,22,23]. In the MFIM system, the VMPs are derived by separating the impedance into its resistance (R-VMP) and reactance (X-VMP) components, while also eliminating any offset. The useful signals are R-VMP and X-VMP. Simultaneous impedance measurement at several (e.g., 3) carrier frequencies improves immunity against electromagnetic interference (EMI) [14,24]. The MFIM system enables the recognition of the type or class of a given vehicle by utilizing comparative classification algorithms [25]. In contrast, MFIM systems are widely applied in fields such as biomeasurements [26], non-destructive testing of metal [27,28,29], metal detection (e.g., manufactured by XP Metal Detectors), impedance spectroscopy [30], and impedance tomography [31], where the measurement of impedance components (resistance and reactance) is critical. Despite its success in the above-mentioned areas, MFIM systems have not yet gained significant recognition or adoption within Intelligent Transportation Systems (ITS) applications. The available literature lacks comprehensive studies on the use of MFIM systems in conjunction with IL sensors for generating resistance R-VMP and reactance X-VMP, which may offer advantages over the more common single-output signal solutions in vehicle classification.

The objectives of the activities presented are as follows: the development of a laboratory test bed (LTB) that enables the testing and validation of the IL sensor system for measuring vehicle parameters at a specified speed. The proposed LTB will be utilized to evaluate the metrological properties of the measuring system under specific laboratory conditions. Additionally, it aims to determine measurement errors across the full range of road speeds, eliminating the need for lengthy and costly on-road experiments, where obtaining a reference speed poses significant challenges. One of our objectives is to first determine and ultimately reduce the influence of vehicle speed on the estimated vehicle parameters across a wide range of speeds. This will be accomplished through software, specifically within the algorithmic layer of the measurement system. The original contribution and novelty of this work lies in the complete design, realization and validation of the LTB, which consists of a scaled model of the vehicle, a scaled model of IL sensors, and a high-accuracy speed controller. The developed LTB enables testing of the measurement system under controlled laboratory conditions, whereas the RTB requires tests to be conducted in uncontrolled road environments. Due to the RTB being located on a road with a speed limit, a thorough investigation of the speed’s influence on VMPs could not be conducted. To develop the LTB, it was essential to obtain physical models of vehicles that interact with IL sensors in the same manner as real vehicles, as well as to design a tool for accurately and consistently setting the passage speeds of these models during measurement trials.

The paper is organized into the following sections. Section 2 introduces the proposed LTB, including a description of the developed components and their properties. Section 3 details the arrangement of the IL sensors, the MFIM system, and the algorithms employed to measure vehicle parameters in motion. Section 4 provides a description of the conducted measurements and presents the obtained results. Section 5 offers a discussion of the findings. Finally, Section 6 summarizes the entire work.

## 2. Equipment for Laboratory Validation

### 2.1. Laboratory Test Bed

The developed LTB (Figure 1) allows the vehicle model to travel at a predetermined speed through the area of operation of the IL sensor field. A rectangular supporting structure, shown in Figure 2, front view, is used as the basis for the LTB.

The LTB generates measurement signals that correspond to those obtained from the RTB, while also considering the modeled EMI produced by the operation of the vehicle’s engine. The vehicle model follows a circular trajectory with a fixed radius. The drive system, which is mounted internally, consists of a motor, planetary gear, an encoder, and a programmable controller. This controller regulates the rotational speed based on data received from the encoder, see Figure 2. The software for the controller allows for the rotation of the arm at a specified rotational speed and for a predetermined number of laps. At the end of the arm, a vehicle model is mounted which is described in detail in the next section. A rotary connector mounted in the motor axis supplies the voltage signal from the generator to the moving vehicle model’s EMI coil. The design parameters of the LTB are listed in Table 1. The vehicle and the IL sensors models are made in the scale 1:50 relative to the real dimensions.

### 2.2. Model of the IL Sensors for LTB

The IL sensor area depicted in Figure 2 features a set of four IL sensors. Due to the adopted circular motion, the RTB flat sensor system was transferred to a segment of the cylinder surface. The target dimensions of the model of four IL sensors are shown in Figure 3a. The design of the carcass on which the IL sensors are mounted was made in the Autodesk Fusion360 software. The plastic carcass was made with 3D printing technology. Several design–3D printing iterations and tests of the sensor preparation technology were carried out. An adjustable mount has been designed to facilitate the correction of the sensor’s position during assembly. The 3D design and the mounting base are shown in Figure 3b. The 3D printing technology ensured the precise production of a solid in accordance with the assumed dimensions. Enameled wire with a thickness of 0.1 mm was used for the winding. The number of turns of each IL sensor was selected in such a way that the sensitivity of the sensor was comparable to that of a real IL sensor. Parameters and measured impedance, using E4980A instrument [32], are shown in Table 2.

The photograph of the completed IL sensor module is displayed in Figure 3c.

### 2.3. Vehicle Numerical and Physical Model

The physical model of the vehicle was developed based on an analysis of its fundamental geometric parameters, including vehicle length, axle spacing, front overhang, and rear overhang. The analysis focused on the causes of visible artifacts in the VMP signals obtained from individual IL sensors. It included numerous physical experiments and simulations of electromagnetic phenomena using the finite element method (FEM) in the ANSYS Maxwell 3D environment.

Figure 4a presents a sample-developed numerical model of the IL sensor and a simplified car chassis, designed in the ANSYS Maxwell 3D environment. This model was used to investigate the obtained R-VMP and X-VMP waveforms (Figure 4b), depending on whether the car’s rim is made of steel or aluminum. Electromagnetic field distributions were calculated for various distances between the vehicle and the sensor (Figure 4c,d).

Based on the obtained field distribution, the average energy stored in the magnetic field of the IL sensor is determined by the following:(1)$$W_{AV}=\frac{1}{4}\int_{V}\vec{\underline{B}}\;\cdot \;{\vec{\underline{H}}}^{★}\;dV$$

Then, the inductance is determined from the following relation:(2)$$L=\frac{4W_{AV}}{I_{peak}^{2}}=\frac{\int_{V}\vec{\underline{B}}\;\cdot \;{\vec{\underline{H}}}^{★}\;dV}{I_{peak}^{2}}$$
where: $\vec{\underline{B}}$ is the complex vector of magnetic induction in space *V*, ${\vec{\underline{H}}}^{★}$ is the complex conjugate vector of magnetic field strength, $I_{peak}$ is the IL sensor supply current amplitude.

The real part of the impedance (resistance) can be determined from the general relation:(3)$$R=\frac{P}{I_{RMS}^{2}}$$

However, for a more accurate representation of reality, it would be necessary to consider the total power loss in the sensor, in the metal object in which eddy currents flow (Equation 4), as well as the radiated power [33].
(4)$$P=\frac{1}{2\sigma }\int_{V}\underline{\vec{J}}\;\cdot \;{\vec{\underline{J}}}^{★}dV$$
where: ${\vec{\underline{J}}}^{★}$ is the complex conjugate current density vector, σ is the conductivity specific of the material in S/m.

In the FEM simulation depicted in Figure 4a, the position of the IL sensor is varied relative to the simplified vehicle model. The findings from these FEM analyses facilitate the development of various simplified vehicle chassis models, which are summarized as follows.

(1)A flat sheet of aluminum effectively simulates the “vehicle body”. This is particularly evident in the R-VMP signal shown in Figure 4b, where the R-VMP level significantly increases when metal components of the vehicle chassis are positioned within the induction field $\vec{\underline{B}}$ of the IL sensor. Corresponding changes in the X-VMP signal are also observed during the interaction between the IL sensor field and the vehicle chassis.(2)Metal elements, such as steel bars or pipe fragments (ring) aligned with the vehicle axles and mounted to the aluminum sheet, generate local extremes in the R-VMP and X-VMP signals, analogous to the behavior observed from the vehicle wheels under real conditions.(3)The intensity of the observed phenomena varies depending on the type of metal used for the rings, as illustrated by the distributions of the magnetic induction vector $\vec{\underline{B}}$ and the current density vector $\vec{\underline{J}}$. In the FEM simulation, the geometric and material parameters were selected to clearly demonstrate two key phenomena: eddy currents and the ferromagnetic core effect. Specifically, ring $A_{1}$ was modeled from ferromagnetic steel, while ring $A_{2}$ was constructed from aluminum (Figure 4a).(4)Artifacts of interest in the R-VMP and X-VMP signals are indicated by vertical dashed lines labeled $A_{1}$ and $A_{2}$ in Figure 4b, which define specific IL sensor positions relative to the vehicle model.(5)The presence of a steel rim and tire belt results in a distinct local maximum in both R-VMP and X-VMP signals. In the case of $A_{1}$, a clear ferromagnetic core effect is observed. The inductance *L* of the IL sensor, as expressed by (Equation 2), is related to the reactance X=2πfL. The local increase in induction $\vec{\underline{B}}$ within the ferromagnetic wheel elements corresponds to a local maximum in the X-VMP signal. Figure 4c illustrates the distribution of induction $\vec{\underline{B}}$ at the cross-section marked as $A_{1}$, showing a notable local maximum. Eddy currents induced in the conducting ferromagnetic materials contribute to eddy current losses, which correlate with an increase in R-VMP alongside X-VMP. The distribution of the current density vector $\vec{\underline{J}}$ is derived from the FEM simulation, revealing that eddy current losses are most pronounced at position $A_{1}$. According to (Equation 3), the resistance increases as the power of losses *P* rises.(6)For ring $A_{2}$, modeled from aluminum (a paramagnetic material), we present hypothetical changes in the R-VMP and X-VMP for an aluminum rim devoid of a typical ferromagnetic belt. When the IL sensor interacts with the aluminum ring at position $A_{2}$, a local minimum in X-VMP and a local maximum in R-VMP are observed. This behavior is attributed to intense eddy current induction effects, which locally diminish the induction $\vec{\underline{B}}$ in the aluminum ring, as shown in the cross-section in Figure 4d. The penetration depth of induction $\vec{\underline{B}}$ is restricted within the aluminum, leading to a reduction in the average energy stored in the magnetic field described by $\vec{\underline{B}}$ and $\vec{\underline{H}}$, as expressed in (Equation 1). This energy is proportional to inductance *L* (Equation 2). Consequently, the high intensity of the eddy currents manifests as a local minimum in X-VMP. The induced current within the aluminum ring, represented by the current density vector $\vec{\underline{B}}$, generates power losses *P* proportional to the resistance *R*. Thus, while we observe a local maximum at $A_{2}$ in R-VMP, it is less intense than the local maximum seen in $A_{1}$.(7)It is worth noting that the occurrence of $A_{1}$ is typical in practice, given the prevalent use of steel-belted tires. The interplay between eddy current phenomena in the aluminum rim and the ferromagnetic core effect may lead to a balance, resulting in a flattened X-VMP signal (indicating no local minima or maxima), while a clear peak appears in the R-VMP signal. This phenomenon highlights why conditioning systems for IL sensors that exclusively provide X-VMP—common in current applications—are inadequate for effective axle detection.

The results of the FEM analyses led to conclusions that enabled the development of a simplified physical vehicle model. Due to the adopted circular motion, the chassis model was transferred to the surface of a cylinder with a constant radius (Figure 5).

The plastic base of the vehicle chassis model was made using 3D printing technology. A rectangular sheet of 1 mm thick aluminum with dimensions proportional to the dimensions of the vehicle as seen from the top was attached to the base. In order to map the shape of the signals obtained from the IL sensors into the space between the axles, a second layer of a smaller sheet of the same thickness was added. To make wheel artifacts visible in the VMPs, ferromagnetic wire rods with a diameter of 2 mm were installed in place of the axles. An EMI coil was mounted in the front part of the model. The terminals of the EMI coil were brought out to the stationary part through a rotary joint. The EMI coil can be powered by a laboratory voltage signal generator. This makes it possible to carry out tests on the system for measuring vehicle parameters in motion in the presence of the EMI interference that arises during the operation of a real vehicle engine.

### 2.4. Setting the Speed

The value of the set tangential speed *v* in the movement of the arm along a circle of radius *R* (Table 1 and Figure 2) is set by the appropriate number of pulses generated by the encoder per second $n_{pps}$ expressed in pps (pulse per second), and can be represented by the relationship:(5)$$v=\frac{n_{pps}}{n_{ppr}}\;\cdot \;2\pi R\;\cdot \;m_{s}\;\cdot \;3.6\;\cdot \;50\;\left(\mathrm{km}/h\right)$$
where $n_{ppr}$ is the number of pulses per one lap of the encoder, expressed in ppr (pulse per rotation); *R* is an arm radius measured from the axis of rotation to the center of the rods imitating the axles in the vehicle model; $m_{s}$ is planetary gear ratio; multiplier of 3.6 is used to convert the speed value from m/s to km/h and multiplier 50 is used to scale the speed in accordance with the scale of the built model of the LTB.

Substituting to (5) the assumed values of $n_{ppr}$ = 10,000 and $m_{s}$ = 0.1 (Table 1) we obtain the relationship for calculation of the tangential speed expressed in km/h:(6)$$v=360\;\cdot \;\pi \;\cdot \;\frac{n_{pps}}{100,000}\;\cdot \;R$$

Based on (Equation 6) the absolute limit value of the tangential speed setting uncertainty can be determined by the following relationship:(7)$$\Delta v=\pm \left(\frac{360\;\cdot \;\pi \;\cdot \;R}{100,000}\;\cdot \;\Delta n_{pps}+\frac{360\;\cdot \;\pi \;\cdot \;n_{pps}}{100,000}\Delta R\right)$$

The $\Delta n_{pps}$ is the uncertainty of realizing the set number of $n_{pps}$ pulses per second generated by the encoder in the Easy-Servo automatic drive control system and equals $\Delta n_{pps}$ = ±1 pps. The uncertainty of the rotation radius ΔR = ±0.0015 m is determined by the uncertainty of the position of the vehicle model axis relative to the arm rotation axis. Based on (Equation 6) and (Equation 7) the uncertainty value of determining the tangential speed *v* for *v* = 10 km/h, and $n_{pps}$ = 2871 pps is as follows:(8)$$\Delta v_{10}=\pm \left(0.0035+0.0487\right)\;\mathrm{km}/h=\pm 0.052\;\mathrm{km}/h$$
and the uncertainty for *v* = 150 km/h, and $n_{pps}$ = 43061 pps is as follows:(9)$$\Delta v_{150}=\pm \left(0.0035+0.731\right)\;\mathrm{km}/h=\pm 0.73\;\mathrm{km}/h$$

On the basis of the second component in (Equation 7), it can be seen that the uncertainty ΔR of determining the radius has a large impact on the uncertainty of setting the speed, especially for the increasing value of the speed, because then the value of the number of set impulses $n_{pps}$ increases linearly with the speed (as shown in examples (Equation 8) and (Equation 9)).

## 3. MFIM System with IL Sensors for Vehicle Parameters Measurement

The laboratory-tested measurement system uses original IL sensors layout, hardware and software solutions.

### 3.1. Hardware Description

The RTB measurement system contains four IL sensors, see Figure 6, mounted in the road as follows: two wide IL sensors (with dimensions 1 m by 2 m), and two slim IL sensors (with dimensions 0.1 m by 3.2 m) arranged alternately starting from the wide IL sensor [21,23]. Wide IL sensors provide a sufficiently high sensitivity for vehicle detection and enable accurate speed measurement and reliable registration of full (not cropped) VMPs. Wide IL sensors are also used for the vehicle body length estimation. Slim IL sensors are used to detect the axle, to measure the wheelbase, and for overhang estimation. The measurement system block diagram is shown in Figure 7.

Each of the four IL sensors works with an analog circuit including an auto-balancing bridge (ABB). The ABB is the front end between the IL sensors and the digital part of the system. Signals from four digital- to-analog converters (DACs) are connected to each of the four inputs of the ABB systems to excite the measuring current. Eight analog-to-digital converters (ADCs) are used to measure the voltages at the outputs of four ABBs. Synchronously clocked DACs, ADCs and FPGA are all parts of the PXI computer.

### 3.2. Software Overview

The software operating on the FPGA implements a direct digital synthesizer (DDS) and, through the DACs, generates multi-frequency excitation for the ABB, thereby stimulating the current in all IL sensors. DDS operates at a sampling rate of 400 kHz. On the basis of the supplied voltage samples from the ADC, the complex voltage values at the output of ABB frontend are calculated by a vector voltmeter (VVM) [21,23,34]. ADCs operate synchronously with DDS at a sampling rate of 400 kHz. The VVM works in the same way as the multi-frequency Lock-in Amplifier [35]. The low-pass filter used in VVM is characterized by a perfectly flat-top passband [36,37].

The measurements, for each IL, are taken simultaneously at three excitation frequencies listed in Table 3. Each VVM provides three 1 kS/s complex voltages (*V*) related to the given IL sensor and three 1 kS/s complex currents (*I*) flowing through that sensor. Next, three 1 kS/s impedances Z=V/I=R+jX are calculated for every IL sensor separately for each of the three excitation frequencies. This means that the MFIM system evaluates 12 impedance signals for four IL sensors simultaneously. The frequencies are selected in such a way that even in the presence of EMI, the redundancy of information allows for the correct measurement of vehicle parameters.

#### 3.2.1. Vehicle Detection and VMPs Extraction

An analysis of changes in the impedance *Z* signals caused by the presence of the vehicle driving through the IL sensor field is used to detect a vehicle. The vehicle is detected if the geometric sum of *Z* exceeds a certain predefined level. The vehicle detection status starts to transfer buffered data into a file. The record contains some pre-event data preceding the vehicle detection (pre-trigger), the main event data, and some post-event data succeeding the end of the vehicle detection status (post-trigger).

The part of the impedance signal containing changes in resistance and reactance parameters (R and X) caused by the driving vehicle is a VMP. The *R* and *X* signals contain a DC component equal in value to the nominal resistance and reactance. This offset is removed from the VMP by subtraction of the signal level before the vehicle presence, which is obtained from the pre-trigger part of the impedance signal. The exemplary extracted VMPs of a passenger car are shown in Figure 8.

#### 3.2.2. Speed Measurement

The calculation of vehicle speed involves analyzing the temporal offset between signal patterns detected by two X-VMP sensors (IL1 and IL3) positioned 1.5 m apart, as shown in Figure 6. These X-VMP signals are preferred due to their increased resistance to electromagnetic interference (EMI), making them more reliable than R-VMPs. To estimate the time delay, cross-correlation of the sensor outputs is performed after each signal has been scaled by its peak amplitude [8]. The time delay corresponds to the point where the cross-correlation function reaches its maximum value [38].

#### 3.2.3. Conversion of VMP Signals from Time to Distance Domain and Synchronization

In the measurement system, the IL sensors are positioned in a linear sequence, with data captured by VMPs appearing in chronological order (see Figure 8). The fixed known distances between these sensors (see Figure 6) allow for the transformation of the time axis into the axis representing the distance traveled by the vehicle. For this transformation, the distance axis resolution is set to 1 cm, meaning that the VMP signals are resampled at 1 cm intervals in the distance domain. This process requires resampling the VMP signals in the time domain, where the new time sample vector is determined by $t_{r}=0.01/$speed. As a result, when this vector is multiplied by the vehicle’s speed, it produces a distance vector with a resolution of 1 cm. For vehicle speeds below 36 km/h, the algorithm performs downsampling, whereas for speeds above 36 km/h, it increases the sampling frequency based on the relation 1/(0.01/(36/3.6))=1 kHz. Once the signals are converted to the distance domain, data from IL1, IL3, and IL4 sensors are shifted by the corresponding mounting distances relative to IL2 (see Figure 6). This allows for the synchronization of VMPs in the distance domain, enabling accurate comparisons of signals recorded at different speeds. Examples of this process are shown in Figure 9.

#### 3.2.4. VMP Analysis and Artifact Detection in the Distance-Domain

The VMP analysis in the distance domain leads to the detection of artifact locations originating from the vehicle front, rear, and axles. The detection of the front and the rear is carried out in order to measure the vehicle’s length. The location of the axles in the body of the vehicle allows us to measure the distance between the axles and the front and rear overhangs. Axle detection employs an Algorithm 1 that enhances the axle information by combining R-VMP and X-VMP obtained from the slim IL sensors [14].

The local maxima in question, originating from the vehicle’s wheel elements in the case of low-suspended vehicles, are very poorly visible and require enhancing. Algorithm 1 is a MATLAB function designed to enhance the axle signal by processing two input signals, *R* and *X*, which represent real and imaginary components of the VMP. The function begins by finding the minimum value of the imaginary signal *X*. This is conducted using the min function, where *x* is the minimum value of *X*, and *k* is the index at which this minimum value occurs. The function then extracts the value of *R* at the same index *k*, corresponding to the minimum value of *X*. The next step is to calculate the gain factor, which is the absolute value of the minimum *X* divided by the corresponding *r* value. A new signal *K* is computed by applying the gain to the real component *R* and then adding the imaginary component *X*. The final step is to normalize the combined signal *K* by dividing it by its maximum value, which ensures that the output signal remains within a standard range. The function works by identifying a key point in the *X* signal and using it to compute a gain factor that is applied to the *R* signal, resulting in an enhanced Axle signal. Exemplary results obtained with Algorithm 1 for a passenger vehicle are shown in Figure 9, where the axle information is visible in the form of peaks.

**Algorithm 1** MATLAB function for enhancing axle signal from *R*, and *X* VMPfunction Axle=AxleEnhance(R,X)[x,k]=min(X)r=R(k)gain=abs(x)/rK=gain·R+XAxle=K/max(K)

Algorithm 1 is applied separately to individual VMPs obtained at different carrier frequencies (Table 3) with slim IL sensors. Moreover, some VMPs contain EMI disturbances, which is why the median is used for the obtained results to mitigate their impact. The detection of the front and the rear of the vehicle is carried out using a software comparator with a specific level of comparison and hysteresis. The comparator comparison level and hysteresis were experimentally selected for the vehicle model.

#### 3.2.5. Vehicle Length Calculation

The locations of the vehicle’s front $D_{f}$ and rear $D_{r}$ found in VMPs (see Figure 9) allows for the calculation of the vehicle length $V_{L}$ in meters as follows:(10)$$V_{L}=D_{r}-D_{f}$$

#### 3.2.6. Wheelbase Calculation

The wheelbase $W_{B}$ of a two-axle vehicle is calculated as follows:(11)$$W_{B}=A_{2}-A_{1}$$
where $A_{1}$ and $A_{2}$ are the detected axle locations in meters.

#### 3.2.7. Overhangs Calculation

The front overhang $O_{f}$ and the rear overhang $O_{r}$ of two-axle vehicle is calculated as follows:(12)$$O_{f}=A_{1}-D_{f}$$
(13)$$O_{r}=D_{r}-A_{2}$$

### 3.3. Compatibility of VMPs from LTB and RTB

The RTB VMP signals of a passenger car are shown in Figure 10, whereas LTB VMPs of a car’s physical model are shown in Figure 11. The measured speed value was used to convert the measurement time axis into the distance traveled by the vehicle during this time. Therefore, the VMPs are represented in terms of the distance traveled by the vehicle relative to the given sensor location. Such a procedure makes it possible to compare the VMP shapes.

A general comparison of the obtained VMPs for a real vehicle and its model comes down to the following conclusions. The sheet of metal in the VMP signals is a good representation of the vehicle body. The ferromagnetic wire acts as a steel belt in the tire, providing artifacts in the form of a local curvature in VMPs. For laboratory testing of the MFIM system, the VMPs compatibility of the model with the real vehicle is sufficient.

## 4. Results

As every vehicle generates EMI [24], a square-wave signal with a fundamental frequency of 14.3 kHz was applied through the EMI-coil (Figure 5) during this test. The interference level was calibrated to mirror the typical levels encountered in real-world traffic conditions. Under these parameters, virtually no interference was detected in the recorded VMPs because the MFIM employs highly selective bandpass filters that are tuned to the operating frequencies specified in Table 3, which feature high stopband attenuation.

For each preset speed ranging from 10 km/h to 150 km/h in increments of 5 km/h, 21 test repetitions were conducted. Thus, a total of 609 runs were made for one full system test. For each run, the MFIM system captured VMPs and calculated the vehicle model speed, length, wheelbase, and rear and front overhangs.

Speed errors for each measurement, computed as the difference between the speed measured by the MFIM system and the preset speed are shown in Figure 12 in the form of a box plot [39]. On each box, the central red mark indicates the median, and the bottom and the top edges of the box indicate the 25th and 75th percentiles, respectively. The whiskers extend to the most extreme data points that are not considered outliers, and the outliers are plotted individually using the red × symbol. The outliers were identified using the MATLAB boxplot function based on the interquartile range (IQR) method.

The measurement results of the wheelbase error and vehicle length error dependent on the preset speed are shown in Figure 13. The values of the front and rear overhang error of the vehicle model in dependence on the preset speed are shown in Figure 14.

The maximum speed measurement error of the vehicle model is 0.895 km/h and occurred at the set speed of 125 km/h. The absolute wheelbase and length measurement errors are 3.5 cm and 4.3 cm, and occurred at 130 km/h and 125 km/h, respectively. The absolute measurement errors of the front and rear overhangs are 4.3 cm and 1.7 cm, which occurred for the set speed of 135 km/h. The maximum speed measurement error of less than 1 km/h demonstrates that the system operates reliably at high speeds. Additionally, the errors in wheelbase and length measurements, which are within a few centimeters, indicate a high level of accuracy. This precision, achieved under laboratory conditions for the MFIM system, is crucial for various applications, including vehicle classification, where even minor discrepancies can result in classification errors. Overall, these results illustrate the effectiveness of the employed methodology in obtaining reliable measurements across a range of speed conditions.

Additionally, the repeatability of the obtained VMPs in the distance domain was evaluated at extreme speeds of 10 km/h and 150 km/h. Figure 15a shows the VMPs in the time domain for two runs of the same vehicle model traveling at a high and a low speed. Both signals were subjected to the same procedure of conversion to the distance domain. Good consistency for the VMP shape was obtained at a large speed range (see Figure 15b).

## 5. Discussion

The LTB results presented in this study were obtained under conditions of constant vehicle speed as the vehicle traversed the IL sensor area. However, variations in speed can be readily observed in the distance domain, where they disrupt the alignment between the VMPs recorded by the wide IL1 and IL3 sensors and the narrower IL2 and IL4 sensors.

In the literature, [9,15,40], the concept of a “magnetic length” of the vehicle is introduced, which differs from the actual vehicle length. The magnetic length is defined as the product of the vehicle’s presence time within the IL sensor field and its speed. While it is possible to estimate the real length from the magnetic length using an experimentally determined correction factor, this coefficient varies between high- and low-suspension vehicles. Employing a single universal correction factor for all vehicles introduces significant errors, with discrepancies of up to several tens of centimeters, as noted in [16].

A further challenge in accurately measuring vehicle length using VMP analysis arises from the fact that a large proportion of modern vehicle components are made of plastic, which does not interact with the IL sensor’s electromagnetic field. Addressing this issue requires a highly selective vehicle classification process based on VMP, supplemented by compensating for these variations using a comprehensive knowledge base of reference vehicle parameters.

The results obtained in this work relied on the careful tuning of the artifact detection algorithms, including setting the appropriate comparator thresholds and hysteresis levels to effectively filter disturbances in the VMP signal. Early vehicle classification is critical during signal processing, as it enables the fine-tuning of the VMP analysis algorithms for more accurate vehicle measurements.

Speed measurement error is the most significant contributor to inaccuracies in vehicle length estimation, particularly due to its impact on the detection of the vehicle’s front end. To mitigate these errors and improve the precision of systems such as the MFIM, it is crucial to implement filters with a flat-top response in the passband, carefully tuned to the vehicle’s speed. This approach preserves the shape of the distance-domain VMP across varying speeds, thus enhancing the repeatability and accuracy of the results.

The results obtained from LTB regarding the repeatability of VMPs obtained at different speeds and converted to the distance domain are of great significance, particularly in the context of vehicle classification. This aspect of our research has direct implications for real-world applications, such as its potential use in traffic management systems where accurate and repeatable vehicle classification can enhance traffic flow analysis and automated control.

## 6. Conclusions

The paper presents the LTB for testing the MFIM system for vehicle parameter estimation with a wide range of speed adjustments. The VMPs measured in the proposed LTB are in good agreement with the ones obtained in RTB, thus the conclusions drawn from LTB testing are also valid for RTB. The proposed LTB enables repetitive runs of the vehicle model through the area of operation of the IL sensor field, and thus it allows to determine measurement errors in a specific range of speed that would be difficult to obtain in RTB. The tests of various IL sensor configurations can easily be conducted in the proposed LTB without the need to rebuild the RTB. The proposed LTB is also equipped with the EMI interference generator and it enables comprehensive analysis of the measurement errors caused by disturbances. The LTB cooperates with the MFIM system in the same manner as the RTB, and it is capable of measuring vehicle speed, wheelbase, length, and overhangs. Therefore, the proposed LTB allowed us to determine the errors for these parameters in a wide range of road velocities from 10 km/h to 150 km/h in steps of 5 km/h. The precision and repeatability of the measurements are demonstrated by analyzing two VMPs in the distance domain at two different speeds, showing consistent results after conversion from the MFIM system. This repeatability is critical to the vehicle classification process based on VMPs in traffic.

## Figures and Tables

**Figure 1 sensors-24-07244-f001:**
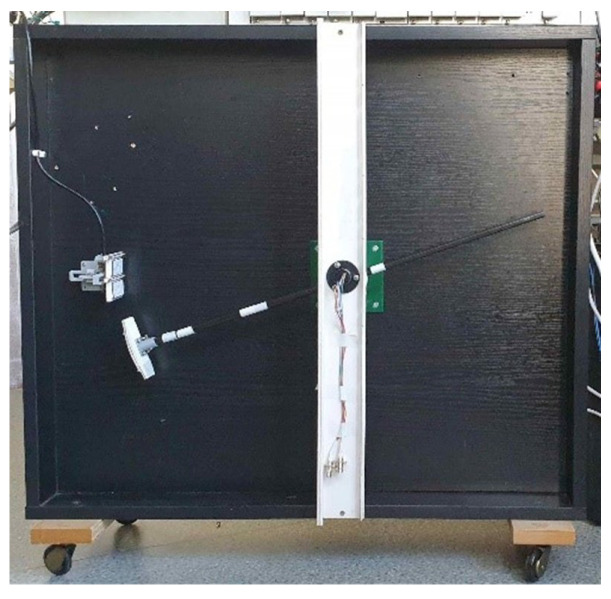
The developed LTB.

**Figure 2 sensors-24-07244-f002:**
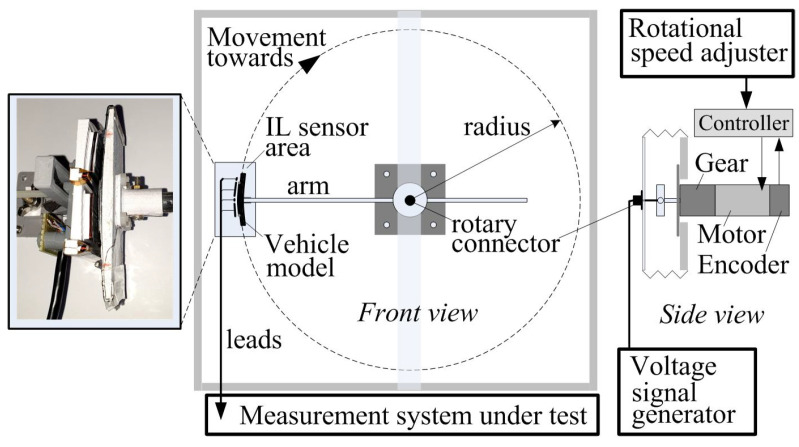
Schematic diagram of the developed LTB.

**Figure 3 sensors-24-07244-f003:**
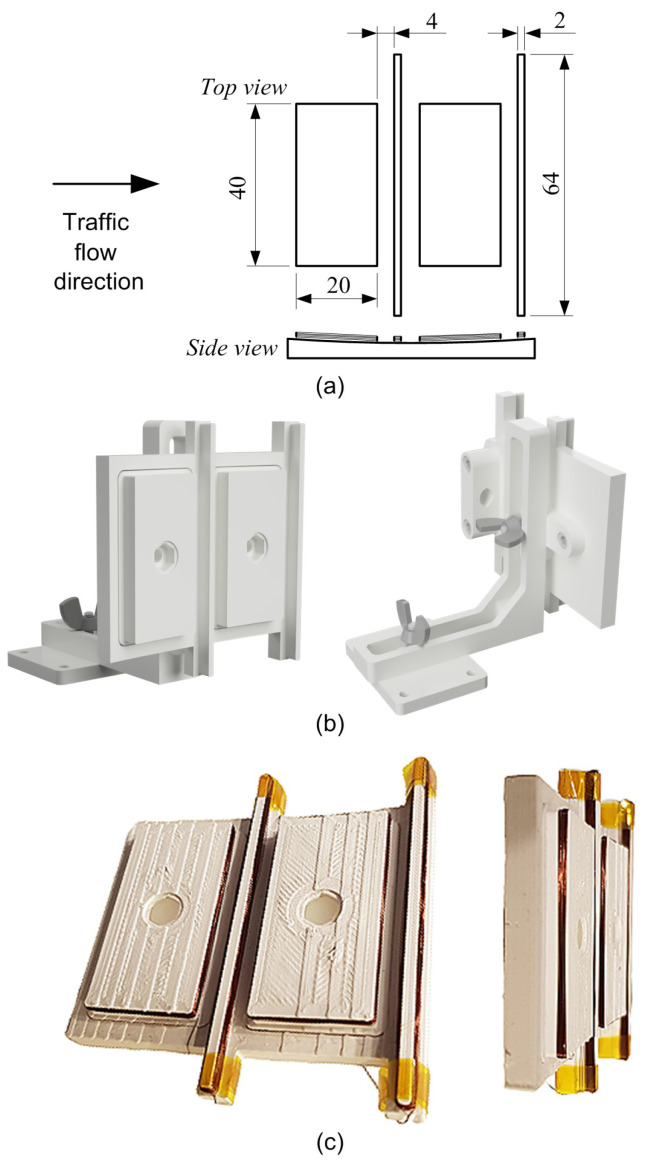
Model of IL sensors: (**a**) dimensions in (mm) and side view of the casing; (**b**) 3D design of the IL sensor core and mount; (**c**) photographs of the produced IL sensors.

**Figure 4 sensors-24-07244-f004:**
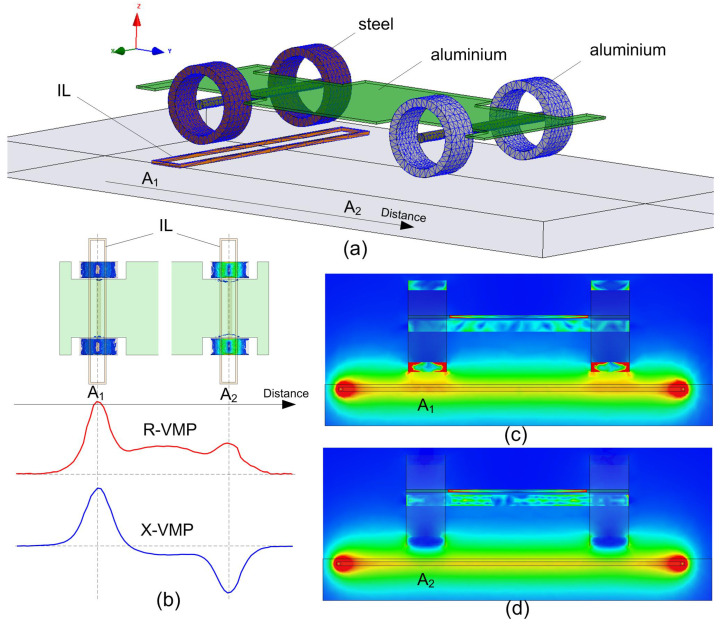
(**a**) FEM model of IL sensor and vehicle, (**b**) example of R-, and X-VMP, (**c**) field cross-section for IL sensor position of $A_{1}$, (**d**) field cross-section for IL sensor position of $A_{2}$.

**Figure 5 sensors-24-07244-f005:**
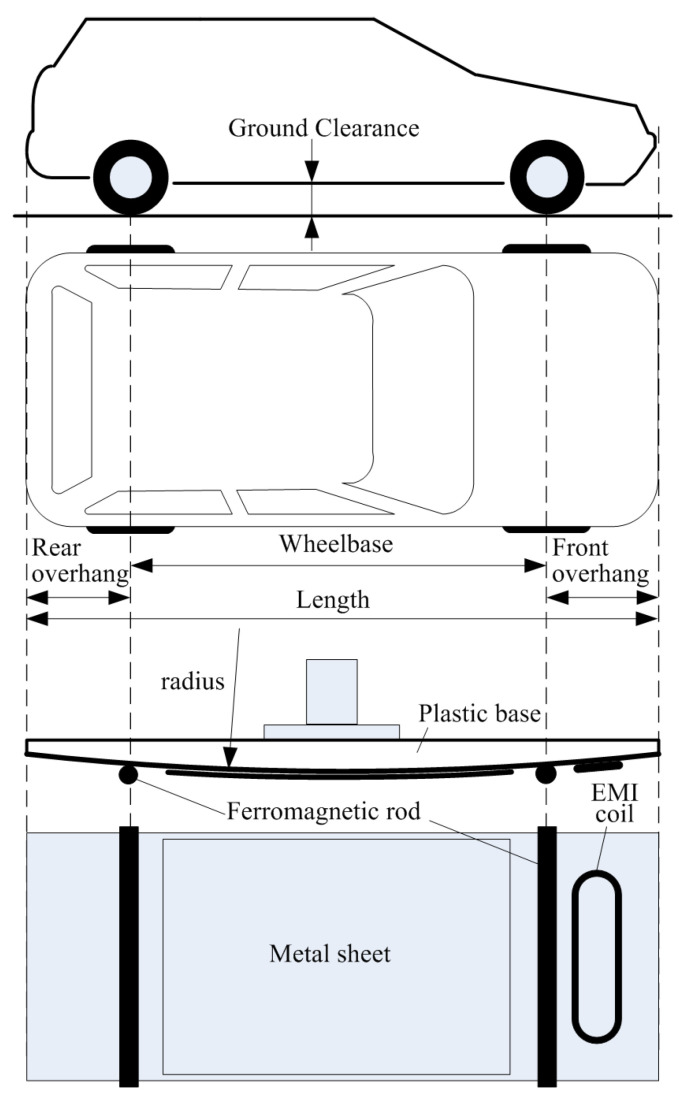
Two-axle physical model of the vehicle.

**Figure 6 sensors-24-07244-f006:**
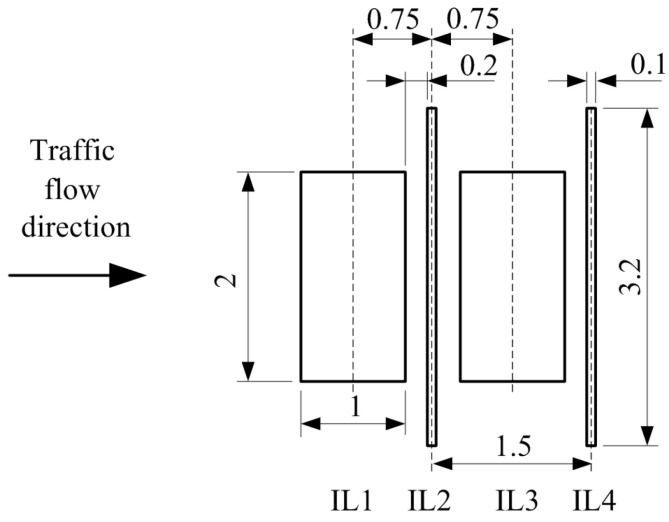
Layout of the RTB IL sensors for the MFIM system. Dimensions are shown in meters.

**Figure 7 sensors-24-07244-f007:**
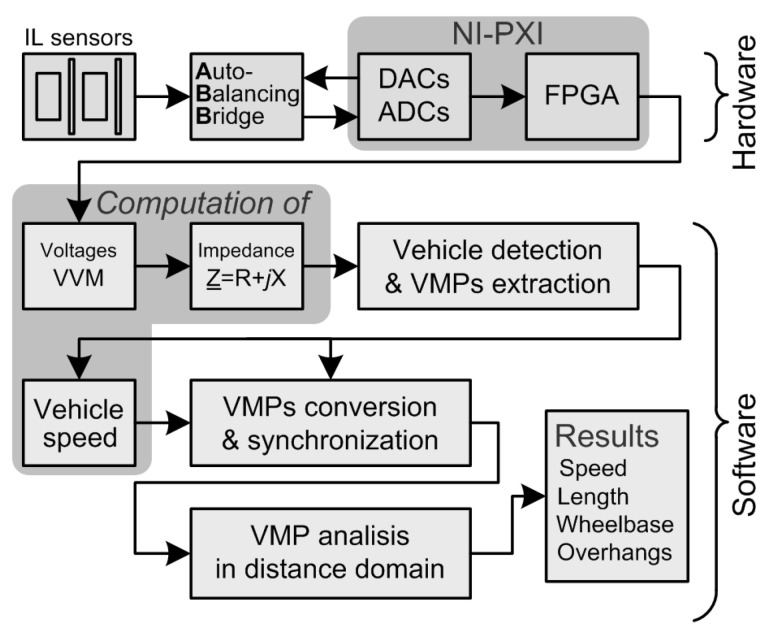
Block diagram of vehicle parameters measurement system.

**Figure 8 sensors-24-07244-f008:**
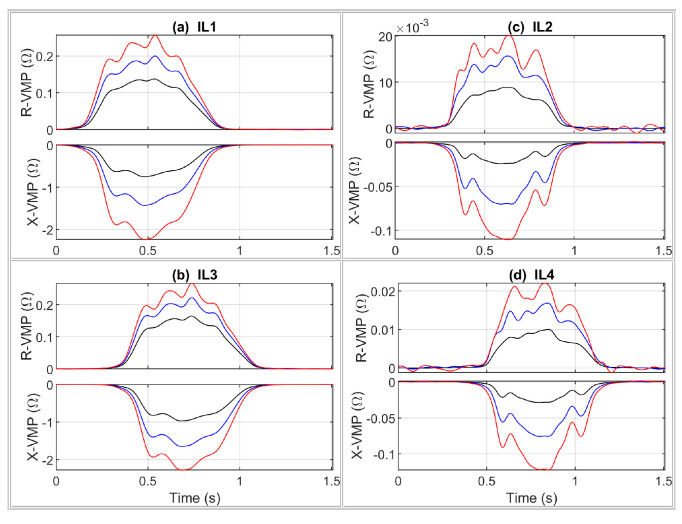
Exemplary extracted VMPs of a passenger car (Toyota RAV4), where R-, and X- means resistance and reactance VMP, respectively. Red, blue, and black VMPs were obtained at the highest, middle, and lowest frequencies, respectively.

**Figure 9 sensors-24-07244-f009:**
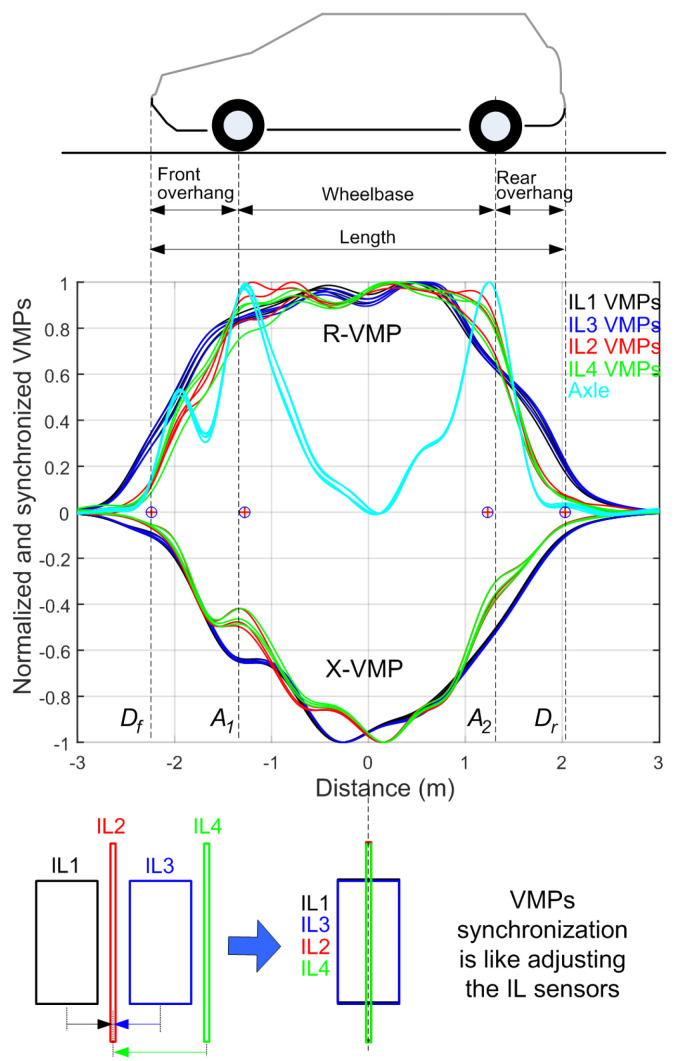
Exemplary normalized VMPs, extracted axle signals and artifacts. Location of detected artifacts ($D_{f}$—vehicle front, $A_{1}$—first axle, $A_{2}$—second axle, $D_{r}$—vehicle rear) are marked with plus circles. Reference values are marked with vertical dashed lines. For each of the four IL sensors, three lines of the same color represent three excitation frequencies, as indicated in Table 3.

**Figure 10 sensors-24-07244-f010:**
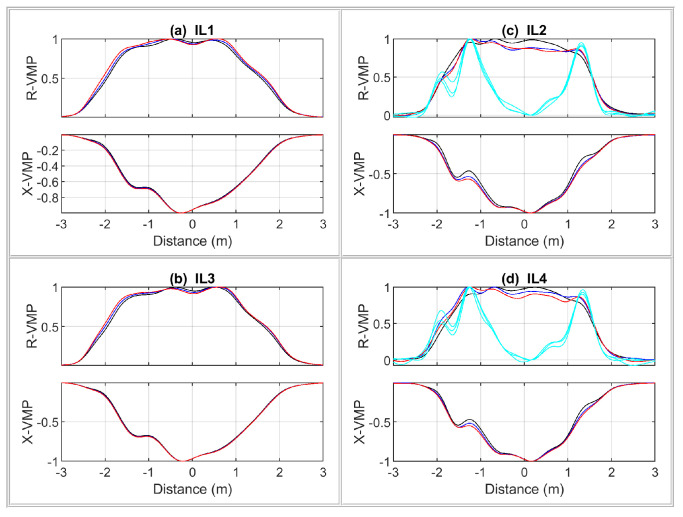
Normalized VMPs and axle signals of the two-axle passenger car (Hyundai i30) registered on RTB.

**Figure 11 sensors-24-07244-f011:**
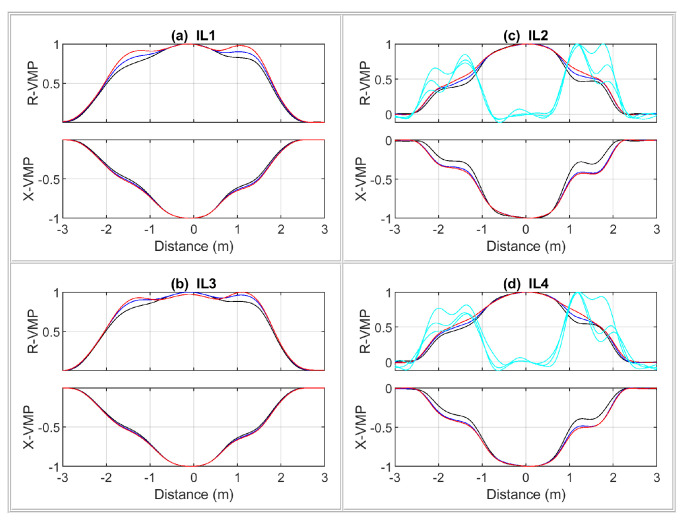
Normalized VMPs and axle signals of physical model of the car registered on LTB.

**Figure 12 sensors-24-07244-f012:**
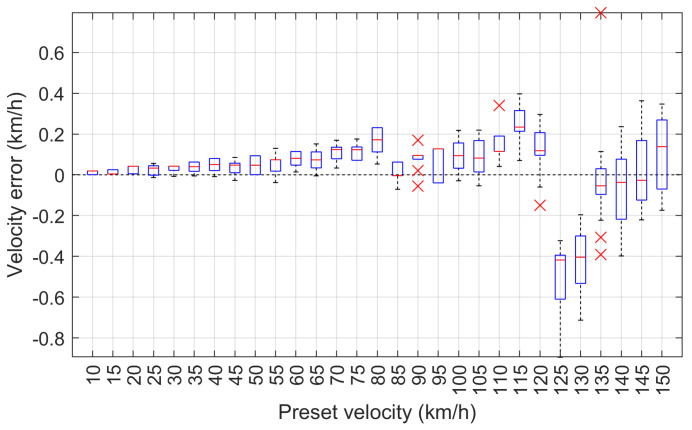
The error of the speed measured by the MFIM system dependent on the preset speed.

**Figure 13 sensors-24-07244-f013:**
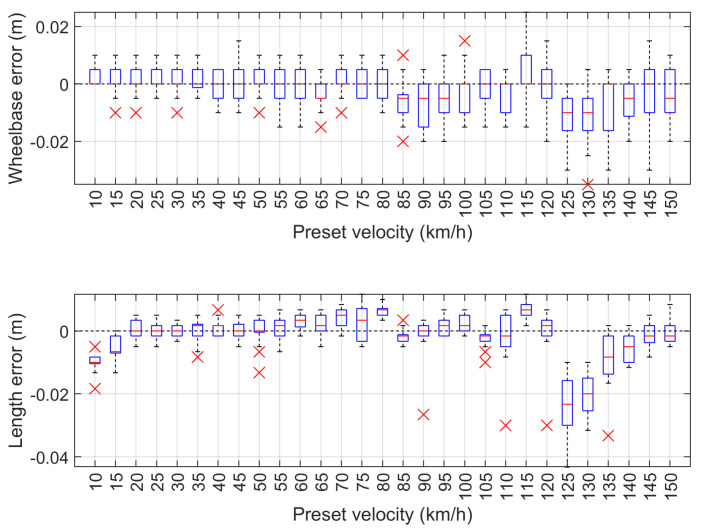
Measurement test results for wheelbase error (**top**) and vehicle length error (**bottom**) dependent on the preset speed.

**Figure 14 sensors-24-07244-f014:**
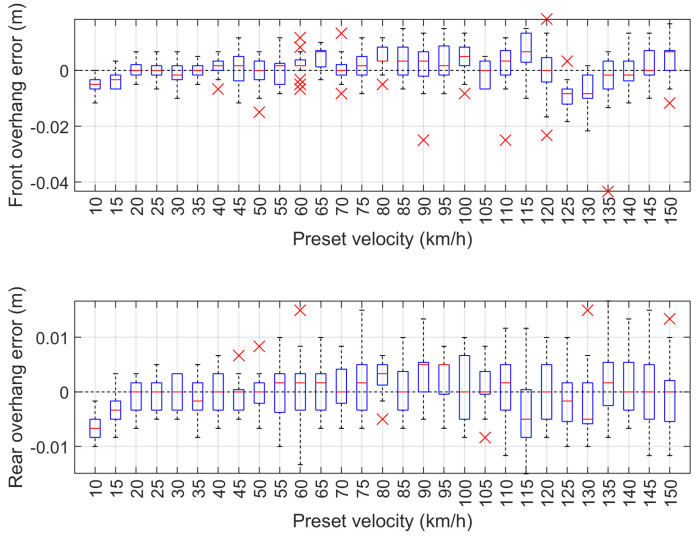
Measurement test results of front overhang error (**top**) and rear overhang error (**bottom**) dependent on the preset speed.

**Figure 15 sensors-24-07244-f015:**
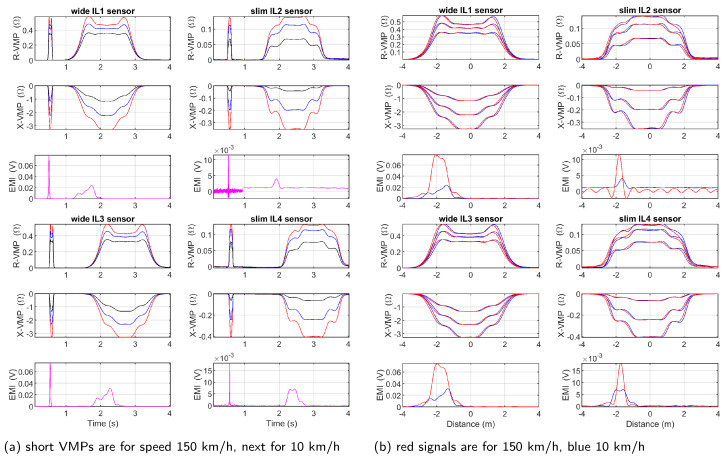
A visual comparison of VMPs obtained at low speed (10 km/h) and high speed (150 km/h) (**a**) in time domain (red, blue, and black VMPs were obtained at the highest, middle, and lowest frequencies, respectively), (**b**) in distance domain after conversion.

**Table 1 sensors-24-07244-t001:** Components and parameters of the LTB.

Parameter or Component	Value or Name
Scale	1:50
Radius	308 mm
Motor	Eazy-Motor
Encoder	10,000 ppr
Controller	Eazy-Servo
Planetary gear	Gear ratio 10:1
Rotary connector	SRC012

**Table 2 sensors-24-07244-t002:** Nominal IL sensor impedance measured at 10 kHz and number of turns.

Sensor	Impedance R+jX (Ω)	Number of Turns
wide IL1	6.5 + *j*1.9	16
slim IL2	16.1 + *j*5.4	31
wide IL3	6.2 + *j*1.7	15
slim IL4	16.4 + *j*5.7	32

**Table 3 sensors-24-07244-t003:** The list of excitation frequencies applied in the MFIM system.

Frequency Value in kHz in a Given Channel	$f_{1}$	$f_{2}$	$f_{3}$
#1: for the first wide IL1 sensor	10	18	27
#3: for the second wide IL3 sensor	13	21	28
#2: for the first slim IL2 sensor	6	15	22
#4: for the second slim IL4 sensor	7	16	24

where: $f_{1}$, $f_{2}$, $f_{3}$—denote excitation frequencies.

## Data Availability

The raw data supporting the conclusions of this article will be made available by the authors on request.

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
