# Peer review of "Validation of Multi-Frequency Inductive-Loop Measurement System for Parameters of Moving Vehicle Based on Laboratory Model"

_sensors, 2024, doi:10.3390/s24227244_

Round 1
Reviewer 1 Report
Comments and Suggestions for Authors
The paper presents a comprehensive study on a multi-frequency inductive-loop (IL) measurement system designed to assess various parameters of moving vehicles under laboratory conditions. The system utilizes four IL sensors to determine vehicle speed, wheelbase, length, and overhangs by analyzing the vehicle's magnetic profile (VMP). The research aims to validate the system's accuracy and reliability within a controlled laboratory test-bed (LTB) before deploying it in real-world scenarios.
Here are some comments for the authors consideration:
(1) While the paper references previous works, a more extensive literature review could enhance the paper by placing the current research within the context of existing studies and highlighting its novel contributions.
(2)The paper could benefit from a more detailed discussion on the validation of the algorithms used for VMP analysis, particularly in the presence of EMI disturbances.
(3) Although the paper presents error analysis, additional data on the precision and repeatability of the measurements would strengthen the conclusions drawn from the study.
(4)The paper could discuss potential real-world applications and implications of the research, such as its use in traffic management systems or vehicle classification.
Comments on the Quality of English LanguageThe quality of the English language in the research paper is generally good, with clear and concise communication of complex technical details. However, there are a few areas where the manuscript could be improved to enhance readability and clarity.
Reviewer 2 Report
Comments and Suggestions for Authors
The paper describes a laboratory test system designed for testing and validation of inductive loop-based sensors. It consists of a car model mounted on a swing arm and a fixed sensor, which consists of a set of inductive loops. The system is designed in 1:50 scale and allows measurements to be made at different vehicle movement speeds (up to 150 km/h when converted to real vehicle speed). The developed system was validated by comparing measurements of the model with the real car measurements. Subsequently, measurements were performed for vehicle movement speeds from 10 to 150 km/h. From the measurements, different parameters were obtained, which were compared with the real values.
The main advantage of the test system is the ability to easily verify the properties of different configurations of induction loops scaled to a given scale. I consider the main disadvantage to be virtually zero variability in vehicle parameters and vehicle travel path. It is not possible to estimate the behavior of the sensor in a realistic environment based on the measurement of a single vehicle model. Although the vehicle model includes an EMI generator, which theoretically can model different types of disturbances, it has not been used at all in the actual work.
Specific values for the parameters of the induction loops used, in particular the number of turns, the wire thickness and possibly the inductance, are missing in the paper (109). Similarly, the frequencies used in the MFIM are not given (206). The same is true for the comparison and hysteresis values (270). It is not stated whether the values used are universally applicable or have been designed specifically for the vehicle model. It is not stated what oversampling algorithm was used (239).
In Section 2.3, electromagnetic field simulations for a greatly simplified car model are presented. In (135) the authors state "The results of the FEM analyses led to conclusions that enabled the development of a simplified physical vehicle model." What results do the authors have in mind? Nowhere are they stated.
Relations 1-4 are nowhere used.
The description of Algorithm 1 (255-265) contains only a verbal description of the mathematical relations previously mentioned. The explanation of the essence of the algorithm is missing. Why does the algorithm improve the axle signal?
Figure 9 shows two lines for each colour. It is not indicated whether these are two measurements for different vehicle speeds or for different frequencies or something else.
In Fig.11 the calculated axle signal could also be displayed. It would help to assess the suitability of the developed model, because especially the signals from the narrow sensors (IL2 and IL4) for the vehicle model are quite different from the signals from the real vehicle measurements (Figures 11 and 10 respectively).
In section 4 is not indicated how the outliers were identified.
The references contain a relatively large number of self-citations (10 out of 33). Citing the Box plot is not necessary (ref.32).
Round 2
Reviewer 1 Report
Comments and Suggestions for Authors
All issues have been addressed.
Comments on the Quality of English LanguageThe whole paper could be polished to make a smooth-reading